# Regulation and Dysregulation of Endothelial Permeability during Systemic Inflammation

**DOI:** 10.3390/cells11121935

**Published:** 2022-06-15

**Authors:** Katharina E. M. Hellenthal, Laura Brabenec, Nana-Maria Wagner

**Affiliations:** Department of Anesthesiology, Intensive Care and Pain Medicine, University Hospital Muenster, 48149 Muenster, Germany; katharina.hellenthal@uni-muenster.de (K.E.M.H.); laura.brabenec@ukmuenster.de (L.B.)

**Keywords:** systemic inflammation, endothelium, vascular permeability, capillary leakage, angiopoietin-Tie2, adrenomedullin, procalcitonin

## Abstract

Systemic inflammation can be triggered by infection, surgery, trauma or burns. During systemic inflammation, an overshooting immune response induces tissue damage resulting in organ dysfunction and mortality. Endothelial cells make up the inner lining of all blood vessels and are critically involved in maintaining organ integrity by regulating tissue perfusion. Permeability of the endothelial monolayer is strictly controlled and highly organ-specific, forming continuous, fenestrated and discontinuous capillaries that orchestrate the extravasation of fluids, proteins and solutes to maintain organ homeostasis. In the physiological state, the endothelial barrier is maintained by the glycocalyx, extracellular matrix and intercellular junctions including adherens and tight junctions. As endothelial cells are constantly sensing and responding to the extracellular environment, their activation by inflammatory stimuli promotes a loss of endothelial barrier function, which has been identified as a hallmark of systemic inflammation, leading to tissue edema formation and hypotension and thus, is a key contributor to lethal outcomes. In this review, we provide a comprehensive summary of the major players, such as the angiopoietin-Tie2 signaling axis, adrenomedullin and vascular endothelial (VE-) cadherin, that substantially contribute to the regulation and dysregulation of endothelial permeability during systemic inflammation and elucidate treatment strategies targeting the preservation of vascular integrity.

## 1. Introduction

Acute systemic inflammation can be triggered by conditions like infection, major surgery, trauma or burns and is potentially life-threatening when the body injures its own healthy tissues due to a maladaptive overshooting immune response resulting in organ dysfunction and mortality. The endothelium makes up the inner lining of all blood vessels and plays a key role in orchestrating the body’s response to systemic inflammation. Under physiological conditions, endothelial cells control the extravasation of inflammatory cells into tissues and regulate coagulation and perfusion through their essential role in vasomotor control [1]. They form a barrier playing a central role in controlling permeability and the distribution of water, cells and molecules from the circulation into tissues [1]. In response to tissue demand, endothelial permeability is dynamically regulated and can be increased or reduced. Endothelial permeability greatly differs between different organs and is essential for organ function. For instance, in the central nervous system the endothelial barrier function is tightest and substantially contributes to formation of the blood–brain barrier. It restricts the movement of solutes and macromolecules into the brain parenchyma due to endothelial junction tightness and limited transcytosis, while selective transfer of substrates is provided by endothelial transporters [2]. Similarly, the endothelial barrier is relatively tight in cardiac and skeletal muscle and in the lung through the formation of continuous capillaries. Conversely, transendothelial pores in fenestrated capillaries allow greater permeability in the endocrine glands, liver or kidney, while discontinuous capillaries enable blood cells to cross the endothelial barrier in the spleen or bone marrow [3]. Thus, endothelial cells substantially contribute to organ-specific homeostasis in the physiological state. Endothelial cells are an active and heterogeneous cell population that continuously responds to the extracellular environment. They play a central role in the first line of defense against invading pathogens in the innate immune response, when components of the bacterial wall such as lipopolysaccharide (LPS) activate pattern recognition receptors expressed on the surface of endothelial cells [4] as well as on immune cells, tissue macrophages or monocytes, inducing the initiation of the inflammatory and coagulation cascades [1]. In local infection, endothelial activation is beneficial as it helps in the walling off of pathogens and substantially controls infection.

Vascular leakage has been identified as a key contributor to disease pathophysiology in different entities of inflammation. Sepsis, caused by systemic pathogen infiltration, is associated with one in every five deaths worldwide [5]. During sepsis, an uncontrolled and overshooting immune response triggers a ubiquitous loss of endothelial barrier integrity that promotes global increased endothelial permeability syndrome (GIPS), with edema formation and systemic hypotension that endanger organ perfusion [6]. Recently, the cytokine storm during coronavirus disease 2019 (COVID-19) was identified to substantially impair endothelial barrier function, inducing pulmonary edema, suggesting that severe COVID-19 is an endothelial disease [7]. A systemic inflammatory response is further observed in patients with trauma, burns or after major surgery, due to extensive tissue damage that results in immune system activation. During surgery, contact activation by foreign substances, infusion therapy and ischemia result in cytokine release and complement activation that trigger endothelial barrier disruption leading to postoperative capillary leakage associated with pulmonary dysfunction, acute kidney injury and delirium [8,9]. For example, signs of blood–brain barrier disruption can be detected in MRI scans of patients undergoing cardiac surgery and may contribute to delirium and persistent cognitive impairment [10,11]. Acute systemic inflammation results in a rapid increase in endothelial permeability, whereas chronic inflammation leads to slow vascular remodeling resulting in a leakier endothelial cell phenotype [3]. In chronic airway inflammation due to asthma or chronic bronchitis, blood vessels enlarge, proliferate and exhibit increased mediator sensitivity and permeability. Sustained endothelial leakiness to plasma proteins leads to chronic airway edema formation [12]. Chronic endothelial hyperpermeability due to pathological angiogenesis is also found in tumors and chronic inflammatory diseases, for example psoriasis or rheumatoid arthritis [13]. Endothelial hyperpermeability has thus been identified as a hallmark of disease pathophysiology in acute and chronic inflammation and enhancing the endothelial barrier function is a novel, frontline approach to fight a potential maladaptive host response to inflammation.

This review focusses on the regulation and dysregulation of endothelial permeability during acute systemic inflammation. While a tightly controlled and organ-specific endothelial permeability is required to maintain organ homeostasis, during critical illness caused by sepsis, or other conditions like trauma, burns or major surgery, a ubiquitous loss of the endothelial barrier function is a key contributor to organ dysfunction and mortality. We elucidate how endothelial cells maintain vascular leakage during leukocyte extravasation to a minimum, suggesting that vascular leakage can be therapeutically reduced while sustaining inflammatory responses at sites of infection or injury. This review dissects the evidence for possible pharmacological strategies targeting the preservation of vascular integrity interfering with signaling pathways and biomarkers that play a key role in mediating the maladaptive host response to systemic inflammation. Interfering with the angiopoietin-Tie2 signaling axis, adrenomedullin or vascular endothelial (VE-) cadherin may be a novel frontline approach to limit endothelial hyperpermeability and associated complications.

## 2. Endothelial Barrier Characteristics in Vascular Quiescence and Inflammation

### 2.1. Composition of the Endothelial Barrier

Endothelial cells are crucially involved in controlling vascular permeability, and a baseline permeability to water, small solutes and gases is physiologic. Increases and reductions in endothelial permeability are essential to answer the needs of perfused tissues and thus to maintain organ homeostasis. For instance, endothelial cells react to hypoxic conditions with increased permeability [14]. While passive diffusion is the chief transport mechanism of low molecular weight substances like gases, ions or solutes, the endothelial monolayer presents a barrier to liquid and high molecular weight substances, such as proteins [15]. Forming a barrier that keeps plasma proteins, especially albumin, in the vascular space, the endothelium plays a key role in establishing colloid osmotic pressure, which is responsible for holding water within the circulation, which, in turn, is essential for functional organ perfusion and oxygen supply to tissues. In the physiological state, endothelial barrier function is mainly maintained by extracellular structures, such as the glycocalyx, extracellular matrix and intercellular junctions including adherens junctions and tight junctions [16] (Figure 1). The endothelial glycocalyx is a 200–400 nm thick “sugar coat” covering the luminal surface of blood vessels and forms a mesh-like layer consisting of proteoglycan and glycoproteins [17,18,19]. Proteoglycans are the major component of the glycocalyx, consisting of glycosaminoglycans (GAGs) including heparan sulfate (HS), hyaluronic acid (HA) and chondroitin sulfate (CS). These GAGs are attached to core proteins and receptors, like syndecans, glypicans and CD44 [19]. The endothelial glycocalyx serves as a barrier protecting endothelial cells from the spontaneous adhesion of leukocytes and platelets and the negative charge of the glycocalyx repels red blood cells and macromolecules [20]. The glycocalyx thus limits the passage of macromolecules to the endothelial surface. In contrast to this, neutralizing the negative charge of glycocalyx can induce vascular permeability [21]. Glycocalyx further functions as a fluid shear stress sensor involved in the production of nitric oxide (NO) by endothelial NO synthase (eNOS) [15], which may influence the vascular barrier and vasomotion [22,23]. On the opposite side, there is the extracellular matrix, which may also be involved in regulating endothelial integrity and vascular barrier function. Endothelial cells secrete laminin polymers, which bind to β_1_-integrins [24,25]. Collagen IV polymers produced by endothelial cells, interact with laminin polymers forming a basement membrane of high tensile strength and elasticity [15]. In the physiological state, extracellular matrix promotes signaling pathways that favor cell adhesion over cell proliferation. Neighboring endothelial cells are connected by junctional proteins such as gap junctions (GJs), tight junctions (TJs) and adherens junctions (AJs). Gap junctions directly connect the cytoplasm of contiguous cells, forming a gate that manages the movement of molecules, ions and electrical pulses between cells. Tight junctions are formed by occludins, claudins and junctional adhesion molecules (JAMs). The expression level of occludin is known to correlate with enhanced vascular barrier function. For instance, arteries show an 18-fold greater amount of occludin, when compared with veins and are thus less permeable [26]. The C-terminus of occludin is indirectly linked to adherens junctions via the zona occludens protein ZO-1 and further via the actin cytoskeleton [15]. Adherens junctions are of special importance in controlling endothelial barrier function. Endothelial adherens junctions contain vascular endothelial cadherin (VE-cadherin), which connects neighboring cells through its extracellular domain. Its cytoplasmatic tail binds to p120-catenin via the juxtamembrane domain (JMD) and to β-catenin and plakoglobin via its C-terminal domain (CTD). β-Catenin or plakoglobin are linked to α-catenin, which indirectly connects VE-cadherin to the actin cytoskeleton [15,27]. Several inflammatory mediators, such as histamine, thrombin or vascular endothelial growth factor (VEGF), increase permeability by affecting the adhesion of VE-cadherin [28]. Activation of Src-family tyrosine kinases induces phosphorylation of VE-cadherin and leads to internalization of the molecule, promoting an increase in vascular permeability [29]. Several phosphorylation sites of VE-cadherin have been identified [30]—among these is the tyrosine residue, Y685, which is most crucial for the induction of permeability in response to inflammatory stimuli [31,32]. Cytoskeletal remodeling is a key interface in the regulation of endothelial permeability. Here, the Rho family of GTPases contains small signaling G proteins that play a crucial role in cytoskeletal dynamics and cell adhesion. Guanine exchange factors (GEFs), GTPase-activating proteins (GAPs) and guanine nucleotide dissociation inhibitors (GDIs) regulate distribution between the active GTP-bound and inactive GDP-bound form of Rho GTPases, for example RhoA and Rac1 [33]. These small GTPases function as double-edged swords in modulating endothelial permeability, depending on the upstream stimulus, their activation characteristics and downstream effectors. Inflammatory stimuli induce RhoA activation that promotes generation of radial stress fibers and increased actomyosin contractility, all of which increase endothelial permeability, whereas RhoA can also induce actin polymerization through its effector mDia stabilizing the endothelial barrier [34,35,36,37]. While favoring endothelial barrier maintenance by strengthening assembly of intercellular junctions, formation of lamellipodia and cortical actin bundles, Rac1 also exhibits opposing effects, promoting destabilization of intercellular junctions responding to permeability-inducing stimuli [36,38,39,40,41]. All of the mechanisms mentioned above can contribute to acute vascular hyperpermeability in response to inflammatory stimuli that leads to extravasation of protein-rich plasma into tissues, referred to as exudate [13], and can endanger organ perfusion when occurring ubiquitously during acute systemic inflammation.

### 2.2. Endothelial Permeability and Leukocyte Extravasation

Recent evidence suggests that enhancing the vascular barrier may improve a patient’s outcome in acute systemic inflammation. Leukocytes cross the endothelium to gain access to peripheral tissues and to fight inflammatory foci. Endothelial cells activate leukocytes and guide them to extravasation sites [42]. This is a crucial defense mechanism of the innate immune system against pathogens and tissue injury caused by trauma, burns or major surgery. Leukocytes further proceed from the extravascular space to the vascular lumen, known as reverse transendothelial migration [43,44]. However, for decades, it has been postulated that vascular leakage is necessary for leukocytes to cross the endothelium and is thus a natural consequence in defense against inflammation [45]. Recent evidence has rejected this hypothesis by showing that vascular leakage and leukocyte traffic occur independently in the same blood vessel [46]. For instance, the cytoplasmatic tail of VE-cadherin contains several phosphorylation sites with different distinctive and selective effects on endothelial cell function [30]. While phosphorylation at tyrosine residue 685 induces endothelial permeability in response to inflammatory stimuli, as discussed earlier in this review, dephosphorylation of tyrosine residue 371 in VE-cadherin upon leukocyte–endothelial cell interaction accelerates leukocyte extravasation [32]. Additionally, endothelial cells limit vascular leakage during leukocyte recruitment into tissues, indicating that leukocyte diapedesis does not require vascular leakage (Figure 1). The size of the endothelial pore for leukocyte diapedesis is strictly controlled by the GTPase RhoA, which modulates actin dynamics to form a tight pore around the leukocyte when crossing the endothelium [47]. Furthermore, the endothelium forms an actin-dependent tight-fitting dome encapsulating the leukocyte to limit vascular disruption [48,49]. Besides paracellular diapedesis, leukocytes are able to cross the endothelium transcellularly [42]. In preclinical animal models, enhancement of vascular barrier function was independent from cytokine production, leukocyte recruitment and pathogen clearance, suggesting that vascular leakage is not an inevitable consequence of leukocyte trafficking [50,51,52]. To conclude, preserving vascular barrier integrity is independent from the ongoing inflammatory response and defense against pathogens and tissue injury. Treatment strategies that decrease vascular leakage without compromising the innate immune response are thus a novel, attractive approach to fight systemic inflammation and will be discussed later in this review.

## 3. Modulation of Endothelial Permeability during Systemic Inflammation—From Mechanisms to Targets

### 3.1. Characteristics of Inflammation-Induced Leakage

Almost two millennia ago, Aulus Cornelius Celsus published his work “De Medicina” and postulated the four cardinal signs and fundamentals of inflammation: rubor, calor, dolor and tumor [53,54], which are particularly mediated by alterations of the vascular barrier and are required in local inflammation to eliminate the causative factor. For example, when tissue is damaged by infectious agents or mechanical injury, emerging microorganisms secrete pathogen-associated molecular patterns (PAMPs), while injured tissue releases damage-associated molecular patterns (DAMPs) [42]. These molecules further activate local immune cells to secrete proinflammatory mediators that activate endothelial cells to express chemoattractants and adhesion molecules on their luminal side [55]. Following this, leukocytes and endothelial cells interact through ligands, leading to adhesion and the para- and trans-cellular migration of leukocytes into inflamed tissue. Besides recruiting leukocytes, the activated endothelium modulates the coagulation cascade, alters vasomotor tone and triggers programmed cell death, leading to compartmentalization of the local inflammatory response [1]. Here, vascular leakage during local inflammation can facilitate blood cell trafficking and extravasation of macromolecules to the site of infection and is thus beneficial and effective in the resolution of inflammation and tissue repair. For instance, extravasated fibrinogen processed to fibrin forms a matrix for the generation of new blood vessels during angiogenesis [56]. Further, the interstitial fluid is collected by lymphatics and transported to lymph nodes, where antigens are presented to the immune system [57]. However, if the inflammatory response is systemic, a ubiquitous loss of endothelial barrier function leads to a maladaptive global increased permeability syndrome (GIPS) with edema formation in the extravascular space that promotes intravascular hypotension due to volume depletion. Owing to fluid accumulation into tissues, the distance for oxygen supply increases and microvascular perfusion is impaired due to a rise in interstitial hydrostatic pressure [58]. Loss of endothelial barrier function has not only been identified as a key contributor to multi-organ dysfunction syndrome (MODS) in sepsis [59], but is also known to influence outcomes after events such as major surgery. Even when avoiding perioperative hypotension and preserving macrohemodynamics [60], there is still a loss of hemodynamic coherence between macro- and micro-circulation, leading to organ failure due to reduced oxygen delivery caused by altered endothelial barrier function. Loss of endothelial barrier function causes organ-specific complications, such as acute respiratory distress syndrome (ARDS) or acute kidney injury (AKI). In the lung, a single layer of endothelial cells establishes a selective barrier to fluids and solutes, the alveolar epithelium is mainly composed by flat alveolar type 1 cells that cover 95% of the alveolar surface [61]. This alveolar–capillary unit forms a tight blood–air barrier enabling gas exchange. During ARDS, the endothelial permeability to liquid, protein, neutrophils and red blood cells increases, inducing inflammation and edema formation in the lung interstitium and fluid translocation into the alveoli that extends the diffusion distance for oxygen and carbon dioxide, and thus substantially impairs gas exchange [61]. During acute kidney injury, loss of endothelial barrier function leads to interstitial fluid accumulation that promotes a volume increase of the kidney causing contusion, stasis and ischemia within the tough fibrous capsula surrounding the kidney [62,63]. Interestingly, the inflammation triggered by acute kidney injury causes distant organ effects compromising the function of other organs, for example the lung [64]. In animal models, kidney injury induces an increase of vascular permeability and edema formation detectable in the lung [65]. Since fluid resuscitation during systemic inflammation further amplifies edema formation and is a major determinant of organ complications, re-establishment of endothelial barrier function is recognized as a frontline approach to improve outcome. In the following we will provide a concise overview of mediators that increase during acute systemic inflammation and are identified to crucially affect endothelial permeability and elucidate novel treatment strategies interfering with these mediators that may finally reverse the lack of urgently needed strategies for the treatment of systemic inflammation.

### 3.2. The Angiopoietin/Tie2 Axis and the Relevance of VE-Cadherin in Protecting Vascular Integrity

The angiopoietin (Ang)-Tie2 signaling axis (Figure 2) is one of the most extensively studied pathways in inducing vascular leakage during systemic inflammation. In 1992, Tie2 was first described as being expressed in endothelial cells [66] and Tie2 knock-out mutation in mice was found to be lethal in utero due to multifocal hemorrhage, diffuse edema formation and impaired vasculogenesis [67]. Tie2 is a transmembrane tyrosine kinase expressed on the endothelium and its ligands angiopoietin 1 (Ang1) and angiopoietin 2 (Ang2) exert different actions on Tie2 despite being highly homologous to each other [68]. Ang1 is a paracrine canonical agonist of Tie2, while Ang2 competitively inhibits Ang1-Tie2 binding in an autocrine manner and thus hinders Tie2 activation. During vascular quiescence, mesenchymal cells secrete Ang1, supporting endothelial survival and vascular stability, while Ang2 is expressed at low levels and co-localizes with von Willebrand factor within Weibel–Palade bodies in the endothelium [41,69]. Conversely, upon stimulation by inflammatory cytokines, Ang2 is secreted by these pre-formed endothelial stores resulting in autocrine Tie2 deactivation [70,71]. Furthermore, endothelial cells shed the Tie1 ectodomain leading to Ang2 binding, resulting in Tie2 antagonism and reducing the agonistic activity of Ang1 [72]. In septic patients’ blood, Ang2 levels are increased, rising to 10- to 200-fold compared with the baseline value within few hours after sepsis onset and correlating with adverse outcome and mortality [73,74]. Conversely, septic patients present low levels of circulating Ang1 [75]. The ratio of circulating Ang2/Ang1 has been described to be even more sensitive and specific in predicting sepsis outcome than either protein alone [76]. During inflammation, there is a maladaptive vicious circle inducing further production of Ang2. In the physiological state, Tie2 is highly activated and suppresses the transcription factor Foxo1 transcribing the Ang2 gene [77]. When Tie2 is antagonized by Ang2 in inflammation, the brake on Foxo1 is released, resulting in de novo synthesis of Tie2. Vascular endothelial protein tyrosine phosphatase (VE-PTP) is a transmembrane phosphatase that forms hetero-oligomers with Tie2, hydrolyzing crucial phosphotyrosines and thus inhibiting Tie2 signaling [78]. Downstream of Tie2 activation in endothelial quiescence, a phosphatidylinositol-3-kinase/protein kinase B (PI3K/Akt) signaling cascade activates Rac1, while also inhibiting RhoA, leading to an increase in cortical actin that strengthens the cytoskeleton [36,79]. Here, PI3K/Akt activates the GTPase-activating protein 1 (IQGAP1) that stabilizes Rac1 in its active GTP-bound form [38]. Conversely, the Rho GTPase-activating protein p190RhoGAP converts RhoA into its inactive state [79]. Further, Tie2 activation results in inhibition of Src kinase, preventing the phosphorylation and internalization of VE-cadherin [37]. These effects are reversed by Tie2 inhibition via Ang2 during inflammation. Additionally, inhibition of Tie2 induces pericyte (cells which wrap around endothelial cells) loss, amplifying microvascular and hemodynamic alterations [80]. Ang2 further induces heparinase secretion leading to enzymatic degradation of the endothelial glycocalyx and contributing to vascular leakage [81]. Besides promoting the loss of endothelial barrier function, Tie2 suppression during inflammation promotes activation of the inflammatory transcription factor nuclear factor kappa B (NF-κB), resulting in the expression of adhesion molecules, such as intercellular adhesion molecule (I-CAM) and vascular cell adhesion molecule (V-CAM) [74], pulling immune cells across the endothelium into inflamed tissue. Beyond sepsis, there are several other conditions, such as influenza, hantavirus, dengue or malaria, or sterile inflammation caused by trauma and major surgery, in which angiopoietin imbalance is indicative of adverse outcome [82,83,84,85,86,87], suggesting that the Ang/Tie2 axis may also display an interface for treatment strategies in other conditions of systemic inflammation. In conclusion, during systemic inflammation, increased Ang2 release and production inhibits Tie2 signaling, leading to loss of endothelial barrier function and expression of endothelial surface adhesion molecules enabling immune cells to migrate into tissues (Figure 2).

A treatment strategy to enhance vascular barrier function during systemic inflammation is to modulate the angiopoietin-Tie2 pathway (Figure 2). One strategy is to attenuate Ang1 expression. For instance, adenovirus-mediated delivery of Ang1 into mice ameliorated hemodynamics and reduced mortality rate upon LPS injection [88]. Similarly, adenoviral delivery of the cartilage oligomeric matrix protein (COMP)-Ang1 protected mice against acute kidney injury in endotoxemia [89]. In a murine model of sepsis induced by cecal ligation and puncture, treatment with recombinant human Ang1 protected mice against organ dysfunction and mortality [90]. Because recombinant human Ang1 has a short half-life in vivo, a more stable variant, matrilin-1-angiopoietin-1 (MAT-Ang1), was used and stabilized the endothelium upon endotoxin administration [91]. The peroxisome proliferator–activated receptor-γ (PPAR-γ) agonist rosiglitazone, originally used as an antidiabetic drug functioning as an insulin sensitizer, has shown to increase levels of Ang1 while lowering the Ang2:Ang1 ratio in malaria, and has thus been tested in clinical trials in pediatric patients with malaria [92,93]. Another therapeutic approach affecting the angiopoietin-Tie2 axis is to limit Ang2 mechanisms responsible for induction of vascular leakage. For instance, small interfering RNA against Ang2 reduced Ang2 expression in murine lungs and improved survival in polymicrobial sepsis induced by cecal ligation and puncture [94]. Additionally, enhancing Tie2 activation may further protect the vascular barrier integrity in systemic inflammation. For example, Tie2 can be activated by interfering with vascular endothelial protein tyrosine phosphatase (VE-PTP), which inhibits Tie2 signaling. Pharmacological inactivation of VE-PTP using the antibody razuprotafib AKB-9778 protected the vascular barrier in a VE-cadherin independent manner [95]. Currently, AKB-9778 is successfully used in clinical trials treating diabetes-induced macula edema [96,97]. Furthermore, a study by Kümpers et al., demonstrated that administration of vasculotide, a synthetic polyethylene glycol (PEG)-clustered Tie2 agonist, reduced murine organ dysfunction and mortality in polymicrobial sepsis [98]. In addition to this, vasculotide was able to rescue mice from influenza virus infection even 72 hours after infection [50]. In their studies, Han and colleagues combined the features of Ang2 inhibition and Tie2 activation, introducing a novel antibody, ABTAA (ANG2-binding and Tie2-activating antibody) [99]. ABTAA triggered Tie2 activation via Ang2 clustering while binding, clustering and activating Tie2. In mice subjected to endotoxin injection, ABTAA suppressed lung edema analyzed by microcomputed tomography, preserved parenchymal integrity in histology and alleviated pericyte loss, induced VE-cadherin tightening, protected against glycocalyx shedding by suppressing heparanase and attenuated inflammation by reducing inflammatory adhesion molecules. Subsequently, ABTAA increased survival upon sepsis induction by cecal ligation and puncture in animals treated with broad-spectrum antibiotics to 70%, when compared to 20% survival in their littermates treated with antibiotics alone. Further, ABTAA increased survival in other sepsis models of endotoxemia and Staphylococcus aureus bacteremia [99]. Thus, interfering with the angiopoietin-Tie2 signaling axis may present a promising treatment strategy with translational relevance for the treatment of patients suffering from systemic inflammation.

### 3.3. Targeting Adrenomedullin Protects the Vascular Barrier

Adrenomedullin (ADM) was first discovered in human pheochromocytoma almost 30 years ago [100]. Originally, the 52-amino-acid-containing, freely circulating peptide was found to only possess vasodilatory properties [101], but to date, it is known to be involved in modulating inflammation, regulation of vascular tone and endothelial barrier function [102]. Mice lacking functional adrenomedullin show lethal hydrops fetalis and cardiovascular abnormalities, indicating that adrenomedullin substantially contributes to development of endothelial barrier function [103,104,105]. Among patients with sepsis and septic shock, increased levels of adrenomedullin were associated with poor outcome and mortality [106,107,108,109,110]. For instance, Marino and colleagues observed a 28-day survival rate of 100% with adrenomedullin levels below 70 pg/mL [107]. Conversely, adrenomedullin infusion reduced endothelial permeability and increased survival in preclinical animal models of sepsis [111,112,113], indicating that adrenomedullin may be a double-edged sword in modulating the host response during sepsis. Adrenomedullin is produced by various cell types including endothelial cells, vascular smooth muscle cells (VSMCs), macrophages and monocytes [114,115,116,117] upon inflammatory stimuli such as interleukin-1 (Il-1), tumor necrosis factor (TNF) or lipopolysaccharide (LPS) [117]. Adrenomedullin then binds to its receptor, consisting of a heterodimer formed by calcitonin receptor-like receptor (CRLR) and receptor activity-modifying protein 2 and 3 (RAMP2, 3) [118,119]. Adrenomedullin has a short half-life of 22 minutes [120] and is removed from the circulation by two distinct mechanisms: after ligation of adrenomedullin with its receptor, the complex is internalized and degraded, whereas proteolytic degradation represents another clearance mechanism for adrenomedullin [121,122,123,124]. Mice lacking the RAMP2 part of the adrenomedullin receptor, as well as adrenomedullin knock-outs, exhibit a phenotype expressing increased vascular permeability and edema formation [125,126]. Conversely, upon staphylococcus aureus toxin exposure, adrenomedullin administration stabilized endothelial barrier function in vitro [127]. Mechanistically, adrenomedullin prevents formation of stress fibers, which pull on intercellular junctions via the cAMP-PKA pathway inducing Rap1 activation and RhoA/ROCK inhibition [128,129]. Here, adrenomedullin acts via its G_s_-coupled receptor, which is interestingly also a target of several mediators that limit endothelial permeability, such as prostacyclin (PGI2), prostaglandin E2 (PGE2) and β-adrenergic agonists [36,111,130,131]. Receptor activation induces an increase in intracellular cyclic adenosine monophosphate (cAMP), facilitated by adenylylcyclase converting adenosine trisphosphate (ATP) into cAMP [132,133]. cAMP elevation promotes activation of Epac1 functioning as a guanine exchange factor for Rap1, which further strengthens the endothelial barrier function in multiple ways [134,135]. Epac1 knock-out mice exhibit increased baseline permeability, suggesting that Epac1/Rap1 activity is crucial for maintenance of endothelial integrity under physiologic conditions [136]. Downstream of its activation, Rap1 inhibits the Rho/ROCK pathway by recruiting and activating RhoGAP Arh-GAP29 to endothelial junctions and thus releases the tension in intracellular radial actin stress fibers [36,137]. Here, Rap1 effector Rasip1 binding to the transmembrane receptor heart of glass (HEG1) is required [138]. In addition to that, Rap1 induces activation of Cdc42 and its effector MRCK by directing FDG5 to intercellular junctions that ultimately increases cortical actin bundling and thus potentiates endothelial barrier strengthening [137]. Cortical actin bundling is further supported by cAMP-induced PKA activation that leads to guanine nucleotide exchange factor Tiam/Vav2-dependent Rac1 activation [36,139,140]. In endothelial cells, knock-out of cortactin lead to increased permeability by interfering with the adrenomedullin pathway and decreasing adrenomedullin secretion, suggesting that cortactin is critically involved in modulating the barrier regulation of adrenomedullin [129]. Further, adrenomedullin triggers vasorelaxation via stimulation of endothelial nitric oxide synthase (eNOS) inducing vasodilation in vascular smooth muscle cells (VSMCs) [141] and directly acts on VSMCs via cAMP-dependent protein kinase A activation resulting in the relaxation of smooth muscle cells [142,143]. For understanding the controversial effects of adrenomedullin on endothelial barrier function and survival of sepsis that have been observed over the past decades, the distribution of adrenomedullin between the circulation and interstitium is crucial (Figure 3). Adrenomedullin present in the circulation protects vascular barrier function in sepsis, whereas adrenomedullin in the interstitium leads to vasodilatation and impairs vascular barrier function, which has become clear when using an antibody inducing a shift of adrenomedullin into the circulation [144]. This will be discussed in the following section when addressing novel therapeutic strategies targeting adrenomedullin that are proposed to preserve vascular barrier function during systemic inflammation.

In preclinical animal models, infusion of adrenomedullin preserved hemodynamics, reduced vascular hyperpermeability and increased sepsis survival [111,112,113]. However, there have been issues hampering the transfer to bedside of adrenomedullin in sepsis. First, systemic administration in higher doses exerts vasodilatory properties, which are detrimental in patients already suffering from severe hypotension in septic shock [145]. Second, adrenomedullin exhibits a short half-life [120,146] and adheres to artificial surfaces used in the clinic [122], suggesting an inconvenient applicability in treatment of septic patients. Adrenomedullin binds to the CRLR/RAMP complex via its C-terminus [147,148]. While complete inhibition of adrenomedullin binding to its receptor did not improve survival in septic mice, treatment with antibodies against the N-terminus of adrenomedullin strongly reduced mortality in mice after sepsis induction by only partial inhibition of adrenomedullin [149,150]. Further, antagonizing the N-terminal side of adrenomedullin with the antibody HAM1101 improved responsiveness to vasopressors and kidney function in murine sepsis [151]. Subsequently, HAM1101 was humanized to HAM8101, named adrecizumab and introduced by Geven et al. [144]. In their studies, pretreatment of rodents with adrecizumab during endotoxin exposure, as well as upon cecal ligation and puncture mimicking sepsis, lead to decreased albumin extravasation into renal tissue, indicating a preserved vascular barrier function and increased murine survival [144]. Interestingly, angiopoietin 1 levels were also augmented in the group treated with adrecizumab [144]. Mechanistically, adrecizumab is a non-blocking monoclonal antibody that binds to the N-terminal side of adrenomedullin and increases its half-life by protecting the N-terminus, where proteolytic degradation occurs and further modulates the equilibrium of adrenomedullin between the blood compartment and interstitium [150] (Figure 3). Given that adrecizumab is a large IgG antibody of 160 kDa, it does not freely diffuse from the circulation into tissues, suggesting that adrecizumab keeps adrenomedullin within circulation. Here, it protects endothelial barrier function, while the vasodilating effect of adrenomedullin on VSMCs can be diminished [150]. In the phase 2 AdrenOSS-2 trial, 300 patients with elevated adrenomedullin (>70 pg/mL) received either adrecizumab or placebo. In this trial, adrecizumab was safe, well tolerated and lead to improvement in organ function and further reduced mortality at day 28 from 28% to 24% [152]. Next, a phase 2b/3 trial, ENCOURAGE, is planned with separate trials for sepsis and septic shock [152]. Conceivably, adrecizumab may display a novel promising approach in biomarker-guided sepsis treatment.

### 3.4. VE-Cadherin as a Target to Seal the Endothelial Barrier

As discussed earlier in this review, VE-cadherin is critically involved in controlling vascular permeability in response to inflammatory stimuli, such as VEGF or histamine [28,32]. Nonetheless, targeting VEGF with a humanized VEGF-neutralizing antibody bevacizumab did not improve survival in experimental sepsis [153]. A pilot study was planned to assess bevacizumab administration in patients with septic shock, however it was withdrawn before patient enrollment. Conversely, London et al., were able to ameliorate vascular leakage in animal models of sepsis and influenza by enhancing localization of VE-cadherin to the endothelial surface in a Robo-Slit-dependent pathway and thus protected mice against a myriad of cytokines playing a role in systemic inflammation [52]. In the early 1990s, Robo was discovered to be involved in axon guidance in Drosophila [154] and later to control oriented cell growth in embryogenesis [155,156,157,158]. In their studies, London et al. showed that activating Robo4 on endothelial cells by recombinant Slit induces enhanced localization of VE-cadherin to the cell membrane and protected mice against vascular leakage in the lung during bacterial endotoxin exposure, polymicrobial sepsis induced by cecal ligation and puncture and H5N1 influenza infection [52]. Thus, they were able to protect the vascular barrier against the hypercytokinemia in systemic inflammation.

Like adrenomedullin, procalcitonin (PCT) belongs to the calcitonin peptide family [159] and is used as an early and predictive sepsis biomarker. In 1993, Assicot and colleagues first described procalcitonin levels to be increased in patients with systemic bacterial infection [160]. Nowadays, procalcitonin can contribute to clinical decision making in septic patients and guide anti-infective therapy in sepsis [161,162,163,164,165]. Furthermore, procalcitonin aids in the therapeutic decision to discontinue antimicrobial therapy as levels decrease when infection is successfully treated [165,166]. In healthy subjects, procalcitonin production by the CALC-I gene is limited to the thyroid C cells and pulmonary neuroendocrine cells, keeping procalcitonin levels to the low picogram range, whereas systemic inflammation induces a ubiquitous expression of procalcitonin in multiple tissues [167]. To date, the exact mechanism of the procalcitonin burst during sepsis remains insufficiently understood, but it is assumed that monocytes adherent to the activated endothelium secrete procalcitonin and further promote its production by various parenchymal cells, such as adipocytes [168]. In preclinical animal models, procalcitonin administration decreased survival, whereas procalcitonin neutralization using antibodies as well as knocking-out procalcitonin production in mice lacking the CALC-I gene augmented macrohemodynamics and significantly decreased mortality, indicating that procalcitonin itself may play a role in sepsis pathophysiology [169,170,171,172,173]. Further, evidence from our research group suggests that procalcitonin directly impairs endothelial cell function and causes cytotoxicity [174,175]. In septic patients’ plasma, procalcitonin exists in two forms, a full-length, 116-amino-acid-long peptide and a 114-amino-acid truncated variant [176,177]. Enzymatic cleavage is facilitated by dipeptidyl peptidase 4 (DPP4) via N-terminal removal of the two amino acids alanine and proline [178]. Procalcitonin exerts actions on the CGRP receptor, which is a heterodimer formed by CRLR/RAMP1 [173,179,180]. Interestingly, elevated procalcitonin levels after major surgery are also indicative for organ dysfunction [181,182]. Recently, data from our research group showed that elevated procalcitonin levels after cardiopulmonary bypass surgery identifies patients with signs of postoperative capillary leakage and increased requirements for fluids and vasopressors, and suggest an impairment of microvascular integrity [180]. We identified that procalcitonin levels of 10 ng/ml, typically observed in septic patients [183], induce severe pulmonary edema formation in healthy wild type mice. Mechanistically, procalcitonin was shown to induce Src-dependent phosphorylation of VE-cadherin at tyrosine 685 and dissociation of adaptor protein p120 from VE-cadherin in endothelial cells (Figure 4). We further verified these findings in vivo, dissecting that mice expressing a non-phosphorylatable Y685F mutant of VE-cadherin were protected against procalcitonin’s edema-inducing effects. We further elaborated that only the truncated 114-amino-acid variant of procalcitonin induces endothelial permeability, whereas inhibition of procalcitonin activation from its longer 116-amino-acid variant protected from capillary leakage. The identification of a direct modulatory role on the endothelial barrier may thus render procalcitonin a novel therapeutic target during systemic inflammation from bacterial infection.

Another novel approach for biomarker-directed treatment of systemic inflammation is to target hyperprocalcitoninemia. In 1998, Nylén et al. first showed that intravenous administration of exogenous procalcitonin doubled the mortality rate in a hamster model of peritoneal sepsis, whereas procalcitonin neutralization with an antiserum improved survival [169]. In a porcine model of polymicrobial sepsis, early and late procalcitonin immunoneutralization showed an amelioration of hemodynamic parameters and a significantly decreased mortality [170,171]. In septic mice infusion of a procalcitonin antibody directed against the N-terminal side led to decreased lung inflammation and improved survival [172]. Further, procalcitonin deficiency increased survival during murine peritoneal sepsis [173]. We and others have shown that blocking procalcitonin receptor by olcegepant, which is a CRLR/RAMP1 antagonist currently used in migraine treatment research [184], is a feasible means to ameliorate procalcitonin’s edema-inducing effects, further protects the vascular barrier function and improves murine survival in polymicrobial sepsis [173,180]. However, recent evidence from the evaluation of olcegepant in porcine sepsis suggests that novel pharmacological substances are needed before targeting the procalcitonin/CRLR signaling axis can be moved towards a clinical evaluation [185]. We dissected that procalcitonin activation depends on truncation by active DPP4, while DPP4 inhibition is able to control procalcitonin effect on the endothelium (Figure 4). Further, DPP4 inhibition by sitagliptin was able to control vascular barrier function in a murine model of polymicrobial sepsis—even 6 hours after disease onset, representing a clinically relevant scenario of treatment time delay. Interestingly, prior studies have also shown that DPP4 inactivation preserves endothelial function and increases murine survival in endotoxic shock [186,187]. In a prospective clinical pilot study of cardiac surgery patients, we shed light on a potential translational relevance as sitagliptin intake prior to surgery was associated with ameliorated postoperative capillary leakage [180]. To date, gliptins have only been approved for oral intake, while septic patients often exhibit gastroparesis and altered bioavailability of orally delivered drugs. However, intravenous administration of sitagliptin was well tolerated in healthy volunteers, protects the vasculature and has cardio- and reno-protective effects [188,189,190]. Additionally, a novel formulation for intravenous DPP4 inhibition is currently under consideration [191], smoothing the way for the transfer from bench to bedside. Accordingly, DPP4 inhibition during systemic inflammation may protect the vascular barrier function by controlling procalcitonin effects and may thus merit clinical evaluation in patients with systemic inflammation.

## 4. Concluding Remarks and Future Perspectives

Endothelial cell activation and loss of endothelial barrier function is a hallmark of the characteristics of overshooting immune responses during systemic inflammation. The crucial contribution of the associated hypotension, edema formation and compromised perfusion to the adverse outcome of patients with systemic inflammation render strategies to preserve endothelial barrier control as frontline approaches to improve patient care. Novel techniques to influence the functionality and integrity of the primary molecular determinants of the vascular barrier and the identification of its modulators during systemic inflammatory diseases will hopefully soon translate into larger randomized controlled clinical trials opening a new era of outcome-relevant treatment options.

## Figures and Tables

**Figure 1 cells-11-01935-f001:**
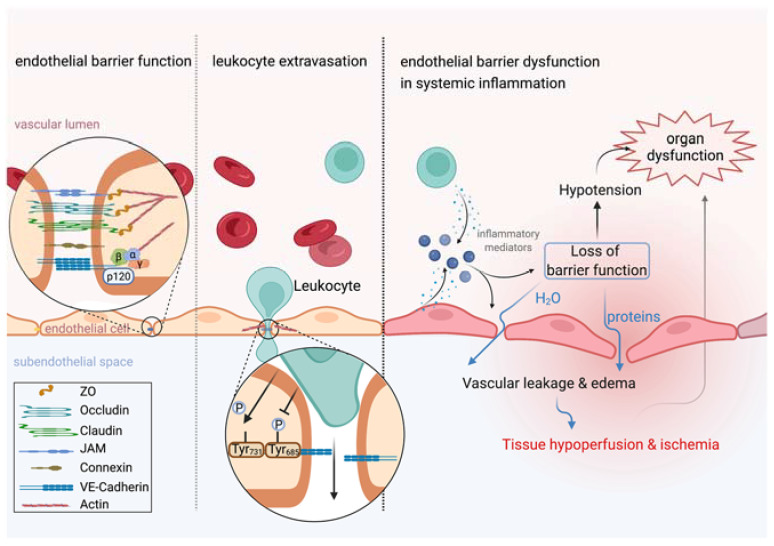
Leukocyte extravasation is independent from vascular leakage. Left: Endothelial barrier function is tightly controlled by intercellular junctions including adherens junctions and tight junctions that connect neighboring endothelial cells. Middle: Leukocyte extravasation is accelerated by dephosphorylation of VE-cadherin at tyrosine residue 371, whereas inflammatory stimuli induce vascular leakage by phosphorylation at tyrosine residue 685. Actin dynamics further contribute to keeping vascular leakage during leukocyte extravasation to a minimum. Right: Inflammation induces rapid release of inflammatory mediators by several cell types, such as endothelial cells or leukocytes. Endothelial cell activation upon inflammatory stimuli, such as VEGF or histamine, induces vascular leakage leading to translocation of protein-rich plasma into the extravascular space. If the inflammatory response becomes systemic, a ubiquitous loss of endothelial barrier function promotes tissue edema formation and hypotension that severely impair perfusion and oxygen supply to tissues and is thus identified as a key contributor to organ dysfunction. This figure was created with BioRender.com.

**Figure 2 cells-11-01935-f002:**
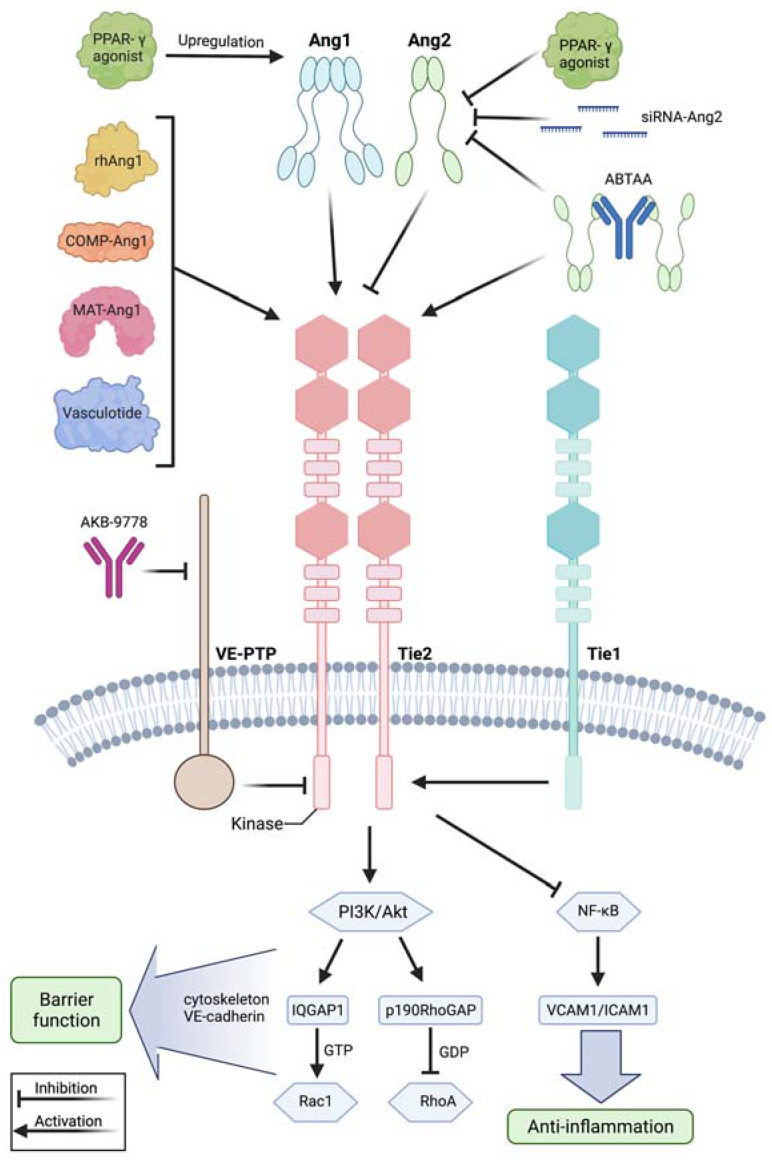
Modulation of the angiopoietin-Tie2 signaling axis during systemic inflammation. Recombinant human angiopoietin 1 (rhAng1), cartilage oligomeric matrix protein (COMP)-angiopoietin 1 (COMP-Ang1), matrilin-1-angiopoietin-1 (MAT-Ang1) and vasculotide are Tie2 agonists with similar action to angiopoietin 1, while peroxisome proliferator–activated receptor-γ (PPAR-γ) agonists upregulate angiopoietin 1 bioavailability. Small interfering RNA (siRNA) against angiopoietin 2 and PPAR-γ agonists reduce angiopoietin 2 expression. AKB-9778 is an antibody directed against vascular endothelial protein tyrosine phosphatase (VE-PTP) and thus indirectly activates Tie2. ABTAA is a novel ANG2-binding and Tie2-activating antibody combining the features of angiopoietin 2 inhibition and Tie2 activation. Tie1 further modulates response at the Tie2 receptor as endothelial cells shed the Tie1 ectodomain leading to Ang2 binding, resulting in Tie2 antagonism and reducing the agonistic activity of Ang1 during inflammation. Mechanistically, these treatment strategies lead to Tie2 receptor agonism, resulting in enhanced vascular barrier function by the PI3K/Akt signaling cascade and anti-inflammation by suppression of transcription factor NF-κB and, thus, of intercellular adhesion molecule (I-CAM) and vascular cell adhesion molecule (V-CAM). Downstream of phosphatidylinositol-3-kinase/protein kinase B (PI3/Akt) activation, there is the GTPase-activating protein 1 (IQGAP1), which activates Rac1 by stabilizing it in its active GTP-bound form, whereas Rho GTPase-activating protein p190RhoGAP converts RhoA into its inactive state. These steps promote an increase in cortical actin that strengthens the cytoskeleton and thus the vascular barrier function, whereas Tie2 activation additionally results in inhibition of Src kinase preventing the phosphorylation and internalization of vascular endothelial-cadherin (VE-cadherin). This figure was created with BioRender.com and adapted from Pariksh SM et al., J Am Soc Nephr 2017 and Wettschureck et al., Physiol Rev 2019 [36,74].

**Figure 3 cells-11-01935-f003:**
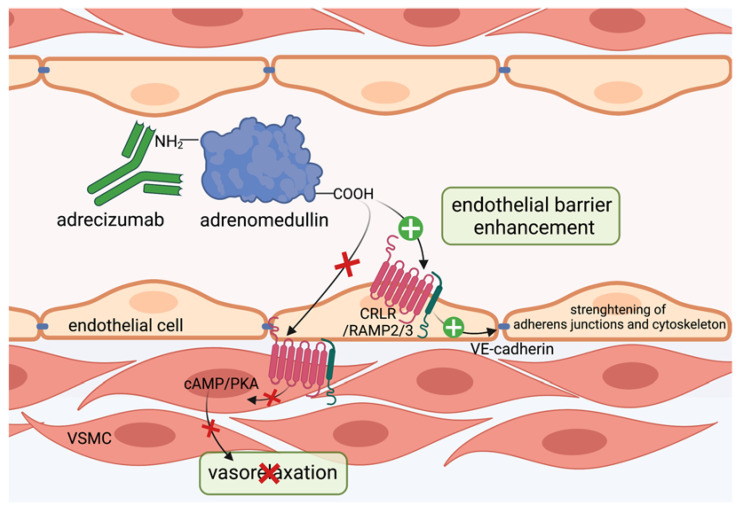
Adrecizumab keeps adrenomedullin within the circulation. The non-blocking monoclonal antibody adrecizumab binds to the N-terminal side of adrenomedullin and modulates the equilibrium of adrenomedullin between the interstitium and blood compartments. It further protects against proteolytic degradation at the N-terminal side and thus increases half-life of adrenomedullin. Kept in the circulation, adrenomedullin protects endothelial cell function via binding to adrenomedullin receptor consisting of the CRLR/RAMP2/3 complex and subsequent stabilization of adherens junctions and cytoskeleton, while vasodilating properties on vascular smooth muscle cells via the cAMP/PKA pathway in the interstitium can be diminished. This figure was created with BioRender.com.

**Figure 4 cells-11-01935-f004:**
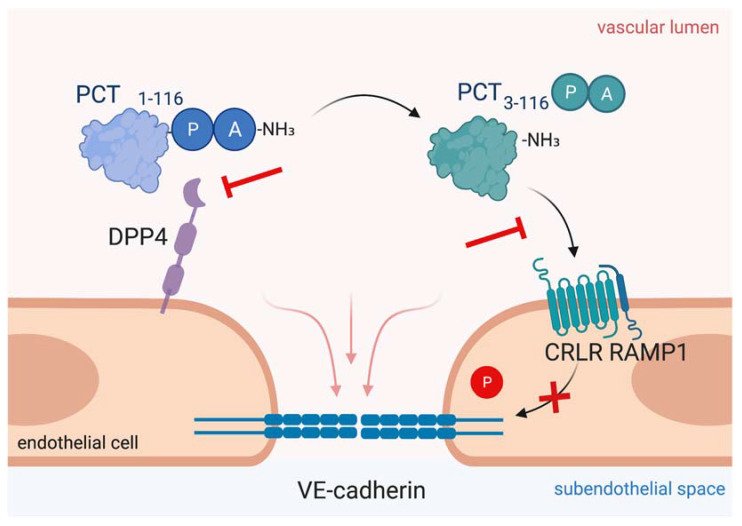
Targeting procalcitonin protects vascular barrier integrity during hyperprocalcitonemia in systemic inflammation. Dipeptidyl peptidase 4 (DPP4) mediates N-terminal truncation of full-length 116-amino-acid-long procalcitonin into its truncated bioactive 114-amino-acid-containing variant. The truncated form binds to CRLR/RAMP1 complex on endothelial cells, which induces phosphorylation of VE-cadherin leading to disruption of VE-cadherin assembly and thus, to vascular leakage induction. As shown in the present figure, antagonizing procalcitonin actions via DPP4 inhibition by sitagliptin and via CRLR/RAMP1 blockage by olcegepant specifically preserves endothelial barrier integrity in murine polymicrobial sepsis. This figure was created with BioRender.com.

## Data Availability

Not applicable.

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
