# Peer review of "Regulation and Dysregulation of Endothelial Permeability during Systemic Inflammation"

_cells, 2022, doi:10.3390/cells11121935_

Round 1
Reviewer 1 Report
Hellenthal et al. summarize recent data reagarding the importance of endothelial permeability with regard to the outcome of systemic inflammation, e. g. sepsis.
Strengths: In general, this review is well organized and provides an important update to the scientific community. It also bridges nicely basic research to emerging therapeutic strategies.
Weaknesses: Although the authors try to explain important processes also mechanistically on the molecular level, they sometimes do not reach that goal and give even misleading informations on the role of the contributing proteins (details, see below). I highly recommend controlling the mechanistic information presented herein against a recent review (https://doi.org/10.1152/physrev.00037.2018) especially dealing with these mechanistic topics.
In the following, I will indicate specific points needing the authors' attentation:
Figure 1:
the text in the enlargements, e. g. Tyr phosphorylation sites on VE-cadherin is not readable. Enlarge resolution or font
Line 210: Celsus is a last name. Please write the first letter in capital. I would like to propose that you use his full name Aulus Cornelius Celsus instead. He is worth it. In addition, the authors should explain why the inflammatory reaction is locally helpful for example at a site of bacterial entry.
Line 262: It must read PI3K/Akt. Please define the abbreviations. Here it is necessary to explain the signaling cascade accurately (see above mentioned review, Fig. 5).
"Activation of Tie2 by angiopoietin-1 supports endothelial cell survival, vessel stability, and endothelial barrier function (243, 432). Tie2 activation protects the mature vasculature from increased permeability induced by VEGF and various inflammatory mediators (160, 166, 527). The angiopoietin-1-induced increase in endothelial barrier function has been shown to involve effects on the actin cytoskeleton by increasing cortical actin through activation of Rac1 and inhibition of RhoA (109, 310) (FIGURE 5). Regulation of Rac1 and RhoA activity through Tie2 involves activation of PI3K and of the IQ domain GTPase-activating protein 1 (IQGAP1), which stabilizes Rac1 in the active GTP-bound form (109), and of the RhoA GTPase-activating protein p190RhoGAP which promotes conversion of RhoA into its inactive state (310)."
If not corrected, this part of the text is misleading, Figure 2 can be used to explain the cascade. All abbrevaitions used should be explained. To my opinion, the upper part of Figure 2 is incorrect. rhAng1, Comp-Ang1 and Mat-Ang1 are externally added agonists on the Tie2 receptor and do not upregulate the endogeneous Ang1 level. This is only done by PPAR-gamma agonists.
Line 362 ff: Also the mechanistic cascade downstream of adrenomedullin is not explained correctly. The activator of Rap1, EPAC1 is not even mentioned and the text is misleading. Please refer to Fig. 6 in the above mentioned review.
"Several mediators, such as adrenomedullin, prostacyclin (PGI2), prostaglandin E2 (PGE2), and β-adrenergic agonists, can limit hyperpermeability induced by inflammatory mediators (140, 207, 265). These mediators have in common that they activate receptors coupled to the heterotrimeric G protein Gs, which mediates activation of adenylyl cyclase, thereby resulting in an increase of intracellular cAMP, which has been shown to reduce vascular permeability (343, 489, 620).
A major mechanism through which cAMP-elevating agents promote endothelial barrier function involves the cAMP effector Epac1, which functions as a guanine exchange factor (GEF) for Rap1 in endothelial cells (103, 155, 248). Activated Rap1 induces increased endothelial barrier function through various mechanisms (378). Rap1 releases radial tension by radial actin stress fibers through inhibition of the Rho/ROCK pathway (19, 589). This effect of Rap1 is mediated by the RhoGAP ArhGAP29, which is recruited to junctions and activated by the Rap1 effectors Radil and Rasip1 (396, 397, 588) (FIGURE 6). In addition, Rap1 stabilizes cell-cell contacts by recruiting the Cdc42-GEF protein FGD5 to junctions (19), which activates Cdc42 and the Cdc42 effector MRCK, leading to increased cortical actin bundling (19). The cAMP-induced Epac1/Rap1 activation can also induce activation of Rac1 via Vav2 (51, 576), and it has also been shown that Rac1 activation in response to cAMP elevation can be mediated by PKA and Tiam1 (50, 51).
Regulation of RhoA/ROCK by the Rap1 effector Rasip1 to decrease stress fiber formation and endothelial permeability appears to require its binding to the transmembrane receptor heart of glass (HEG1) (113) (FIGURE 6). Interestingly, HEG1 also binds Rap1-bound cerebral cavernous malformations 1 protein/Krev interaction trapped protein 1 (CCM1/KRIT1) and thereby recruits it to the endothelial junctions which has also been shown to lead to reduced stress fiber formation and RhoA activity (294, 491). The relevance of the different pathways downstream of Rap1 for the regulation of endothelial barrier function is not clear. However, recently a cooperation of the ARHGAP29-mediated inhibition of RhoA to decrease radial tension and activation of Cdc42 to induce cortical actin bundling has been demonstrated (379).
In addition to Epac1, cAMP promotes endothelial barrier function also through PKA (50, 298). Activation of PKA can promote cortical actin formation by activation of the small GTPase Rac1 through the guanine nucleotide exchange factor Tiam1 and Vav2 (50) or through activation of vasodilator-stimulated phosphoprotein (VASP) (447, 448). It is also well-known that PKA can inhibit the Rho/Rho-kinase pathway (409, 431), a mechanism which might also contribute to stabilizing the endothelial barrier."
Please make use of this information here, by making are more accurate short version and refer to the review.
Author Response
Review Report 1
Response to comments of reviewer #1
Hellenthal et al. summarize recent data reagarding the importance of endothelial permeability with regard to the outcome of systemic inflammation, e. g. sepsis.
Strengths: In general, this review is well organized and provides an important update to the scientific community. It also bridges nicely basic research to emerging therapeutic strategies.
Weaknesses: Although the authors try to explain important processes also mechanistically on the molecular level, they sometimes do not reach that goal and give even misleading informations on the role of the contributing proteins (details, see below). I highly recommend controlling the mechanistic information presented herein against a recent review (https://doi.org/10.1152/physrev.00037.2018) especially dealing with these mechanistic topics.
In the following, I will indicate specific points needing the authors' attentation:
Authors reply. The authors thank this reviewer for his positive comments and for highlighting the strengths of our review. Below, the authors made every effort to address all of the reviewer’s comments in full detail. In the manuscript, all changes are highlighted by use of the “track changes” mode.
Comment #1. Figure 1: the text in the enlargements, e. g. Tyr phosphorylation sites on VE-cadherin is not readable. Enlarge resolution or font
Response to Comment #1. In response to the reviewer’s concern, we increased the font size of the text in figure 1 for better readability in the revised version of the manuscript.
Comment #2. Line 210: Celsus is a last name. Please write the first letter in capital. I would like to propose that you use his full name Aulus Cornelius Celsus instead. He is worth it. In addition, the authors should explain why the inflammatory reaction is locally helpful for example at a site of bacterial entry.
Response to Comment #2. In response to this reviewer’s suggestion, we changed the name to Aulus Cornelius Celsus and provided a more detailed description on the local inflammatory response that is required for effective pathogen clearance and tissue repair from mechanical injury (Pages 5-6/lines 232-250 in the revised version of the manuscript).
Comment #3. Line 262: It must read PI3K/Akt. Please define the abbreviations. Here it is necessary to explain the signaling cascade accurately (see above mentioned review, Fig. 5).
"Activation of Tie2 by angiopoietin-1 supports endothelial cell survival, vessel stability, and endothelial barrier function (243, 432). Tie2 activation protects the mature vasculature from increased permeability induced by VEGF and various inflammatory mediators (160, 166, 527). The angiopoietin-1-induced increase in endothelial barrier function has been shown to involve effects on the actin cytoskeleton by increasing cortical actin through activation of Rac1 and inhibition of RhoA (109, 310) (FIGURE 5). Regulation of Rac1 and RhoA activity through Tie2 involves activation of PI3K and of the IQ domain GTPase-activating protein 1 (IQGAP1), which stabilizes Rac1 in the active GTP-bound form (109), and of the RhoA GTPase-activating protein p190RhoGAP which promotes conversion of RhoA into its inactive state (310)."
If not corrected, this part of the text is misleading, Figure 2 can be used to explain the cascade. All abbrevaitions used should be explained. To my opinion, the upper part of Figure 2 is incorrect. rhAng1, Comp-Ang1 and Mat-Ang1 are externally added agonists on the Tie2 receptor and do not upregulate the endogeneous Ang1 level. This is only done by PPAR-gamma agonists.
Response to Comment #3. We acknowledge this reviewer’s comment that the review would benefit from a more precise description on the PI3K/Akt cascade and adapted our text and figure 2 in the revised version of the manuscript to the information provided by the excellent review from Wettschureck et al. (Page 7, line 315-319 and page 10, line 416-422 of the revised version of the manuscript). In addition, we agree with this reviewer’s concern that rhAng1, COMP-Ang1, MAT-Ang1 are agonists of Tie2 receptor and modified figure 2 accordingly (Page 9).
Comment #4. Line 362 ff: Also the mechanistic cascade downstream of adrenomedullin is not explained correctly. The activator of Rap1, EPAC1 is not even mentioned and the text is misleading. Please refer to Fig. 6 in the above mentioned review.
"Several mediators, such as adrenomedullin, prostacyclin (PGI2), prostaglandin E2 (PGE2), and β-adrenergic agonists, can limit hyperpermeability induced by inflammatory mediators (140, 207, 265). These mediators have in common that they activate receptors coupled to the heterotrimeric G protein Gs, which mediates activation of adenylyl cyclase, thereby resulting in an increase of intracellular cAMP, which has been shown to reduce vascular permeability (343, 489, 620).
A major mechanism through which cAMP-elevating agents promote endothelial barrier function involves the cAMP effector Epac1, which functions as a guanine exchange factor (GEF) for Rap1 in endothelial cells (103, 155, 248). Activated Rap1 induces increased endothelial barrier function through various mechanisms (378). Rap1 releases radial tension by radial actin stress fibers through inhibition of the Rho/ROCK pathway (19, 589). This effect of Rap1 is mediated by the RhoGAP ArhGAP29, which is recruited to junctions and activated by the Rap1 effectors Radil and Rasip1 (396, 397, 588) (FIGURE 6). In addition, Rap1 stabilizes cell-cell contacts by recruiting the Cdc42-GEF protein FGD5 to junctions (19), which activates Cdc42 and the Cdc42 effector MRCK, leading to increased cortical actin bundling (19). The cAMP-induced Epac1/Rap1 activation can also induce activation of Rac1 via Vav2 (51, 576), and it has also been shown that Rac1 activation in response to cAMP elevation can be mediated by PKA and Tiam1 (50, 51).
Regulation of RhoA/ROCK by the Rap1 effector Rasip1 to decrease stress fiber formation and endothelial permeability appears to require its binding to the transmembrane receptor heart of glass (HEG1) (113) (FIGURE 6). Interestingly, HEG1 also binds Rap1-bound cerebral cavernous malformations 1 protein/Krev interaction trapped protein 1 (CCM1/KRIT1) and thereby recruits it to the endothelial junctions which has also been shown to lead to reduced stress fiber formation and RhoA activity (294, 491). The relevance of the different pathways downstream of Rap1 for the regulation of endothelial barrier function is not clear. However, recently a cooperation of the ARHGAP29-mediated inhibition of RhoA to decrease radial tension and activation of Cdc42 to induce cortical actin bundling has been demonstrated (379).
In addition to Epac1, cAMP promotes endothelial barrier function also through PKA (50, 298). Activation of PKA can promote cortical actin formation by activation of the small GTPase Rac1 through the guanine nucleotide exchange factor Tiam1 and Vav2 (50) or through activation of vasodilator-stimulated phosphoprotein (VASP) (447, 448). It is also well-known that PKA can inhibit the Rho/Rho-kinase pathway (409, 431), a mechanism which might also contribute to stabilizing the endothelial barrier."
Please make use of this information here, by making are more accurate short version and refer to the review.
Response to Comment #4. We agree with this reviewer’s concern that a more detailed description on downstream effect of adrenomedullin is needed. We performed an additional literature search and provided information on stabilization of the endothelial barrier by Gs-coupled receptors in the revised version of the manuscript (Pages 10/11, lines 454 to 472). We believe the reader now gets a closer insight into cellular mechanisms being responsible for adrenomedullin actions.
Reviewer 2 Report
This review manuscript by Hellenthal et al summarizes the regulation of endothelial barrier function and its contribution to the development of systemic inflammation. The review is solely focused on angiopoietin-Tie signaling and VE-cadherin-mediated control of endothelial barrier integrity. As such, the overall presentation of key structural elements and mechanisms of EC barrier regulation seems incomplete.
1. Cytoskeletal remodeling is the major modulator of endothelial permeability, yet the authors have failed to discuss this essential phenomenon of endothelial barrier regulation. A brief section describing the role of cytoskeletal reorganization in determining endothelial barrier integrity and role of Rho GTPases in this process will be helpful.
2. The manuscript nicely describes the mechanisms leading to endothelial permeability and inflammation but it fails to provide discussion on the interplay between these two pathological events. It will be better to add a brief description on related sections how increased endothelial permeability contributes to inflammatory responses and vice versa.
3. A description linking endothelial dysfunction (permeability, inflammation) to any particular disease such as ARDS, sepsis, etc. is missing in the manuscript.
4. A thorough revision of the manuscript is suggested to avoid spelling and grammar errors; a few examples are highlighted.
Author Response
Review Report 2
Response to comments of reviewer #2
This review manuscript by Hellenthal et al summarizes the regulation of endothelial barrier function and its contribution to the development of systemic inflammation. The review is solely focused on angiopoietin-Tie signaling and VE-cadherin-mediated control of endothelial barrier integrity. As such, the overall presentation of key structural elements and mechanisms of EC barrier regulation seems incomplete.
Authors reply. We thank this reviewer for his critique and constructive comments on our review. Below, please find a point-by-point response to all of your comments.
Comment #1. Cytoskeletal remodeling is the major modulator of endothelial permeability, yet the authors have failed to discuss this essential phenomenon of endothelial barrier regulation. A brief section describing the role of cytoskeletal reorganization in determining endothelial barrier integrity and role of Rho GTPases in this process will be helpful.
Response to Comment #1. We are grateful for this suggestion and this reviewer was right to concern the missing information on cytoskeletal dynamics regulating endothelial barrier function. In the revised version of the manuscript on page 4, lines 156 to 171, we added a description how the members of the Rho GTPase family modulate endothelial permeability by cytoskeletal remodeling and included new references accordingly.
Comment #2. The manuscript nicely describes the mechanisms leading to endothelial permeability and inflammation but it fails to provide discussion on the interplay between these two pathological events. It will be better to add a brief description on related sections how increased endothelial permeability contributes to inflammatory responses and vice versa.
Response to Comment #2. We acknowledge this reviewer’s comment and encouragement to provide a closer link between endothelial permeability and inflammatory responses. To address this issue, we introduced additional information on how endothelial permeability and the inflammatory response contribute to eliminate the causative factor in local inflammation in the revised version of the manuscript (Pages 5/6, lines 232-250). To the best of our knowledge, vascular leakage is not the inevitable consequence of leukocyte recruitment into inflamed tissues, but rather a maladaptive mechanism and key contributor to lethal outcome, when inflammation becomes systemic (Filewod NC AJRCCM, 2019). We addressed that issue in the chapter about “Endothelial permeability and leukocyte extravasation” (Pages 4/5, lines 177 to 211).
Comment #3. A description linking endothelial dysfunction (permeability, inflammation) to any particular disease such as ARDS, sepsis, etc. is missing in the manuscript.
Response to Comment #3. We appreciate this reviewer’s suggestion and added additional information on organ-specific complications that are closely linked to endothelial dysfunction, with the prominent examples of acute respiratory distress syndrome (ARDS) and acute kidney injury (AKI) and how they influence each other in the revised version of the manuscript (Page 6, lines 261 to 277).
Comment #4. A thorough revision of the manuscript is suggested to avoid spelling and grammar errors; a few examples are highlighted.
Response to Comment #4. We thank this reviewer for the helpful suggestions and corrected spellings in the revised version of the manuscript.
Round 2
Reviewer 2 Report
The points of previous review have been adequately addressed